# Social network characteristics associated with mid-to-older aged adults' co-engagement in physical activity

Martin Anderson[1]*, Srebrenka Letina[1], Mark McCann[1], Jelena Milicev[1], Grace Dibben[1], Abi MacDonald[1], Kirstin Mitchell[1], Laurence Moore[1], Jonathan R. Olsen[1], Victoria J. Palmer[1], Benjamin P. Rigby[2], Sharon A. Simpson[1], Meigan Thomson[1], Emily Long[1]

1 MRC/CSO Social and Public Health Sciences Unit, University of Glasgow, Glasgow, United Kingdom,
2 Population Health Sciences Institute, Newcastle University, Newcastle-upon-Tyne, United Kingdom

* martin.anderson@glasgow.ac.uk

## Abstract

### Objectives

Physical activity is associated with a greater quantity and quality of social connections. Participating in physical activity with others (co-engagement) has dual physical and social benefits that can promote healthy ageing. We aimed to understand the social network characteristics of mid-to-older aged adults associated with co-engagement in physical activity.

### Methods

Adults aged 55–75 years completed a social network survey (3679 social contacts reported by 140 participants). Multilevel modelling was used to identify the characteristics of participants, social contacts, and relationships that were predictive of co-engagement in physical activity.

### Results

Network size and relationship quality were not associated with co-engagement. Similarity in age, greater interaction frequency, closer geographic proximity, and shorter relationship length were associated with higher odds of co-engagement.

### Discussion

For co-engagement, the quality and quantity of relationships were less important than the convenience and accessibility of relationships, particularly newer relationships. As such, co-engagement ties can be understood as part of a dynamic social convoy which fulfil a specific function at a specific life stage. An understanding of naturally

**Data availability statement:** All CSV and R files are available from the figshare database (https://doi.org/10.6084/m9.figshare.27275730.v1)

**Funding:** All authors were employed by the MRC/CSO Social and Public Health Sciences Unit, University of Glasgow, and supported by the Medical Research Council [grant numbers MC_UU_00022/1; MC_UU_00022/2; MC_UU_00022/3; MC_UU_00022/4; and Chief Scientist Office [grant numbers SPHSU16; SPHSU17; SPHSU18; SPHSU19]. The researchers were independent of the funders; the funders had no role in the study design, data collection, analysis and interpretation of data, the decision to publish, or the preparation of the manuscript.

**Competing interests:** The authors have declared that no competing interests exist.

occurring tendencies for co-engagement may be utilised to identify leverage points for the development of interventions.

## Introduction

Increases in life expectancy have resulted in a global rise in both the number and proportion of older people in the population. The World Health Organization (WHO) have predicted that, between 2015 and 2050, the proportion of the world's population aged over 60 years will nearly double, from 12% to 22%, to a population of 2.1 billion people [1]. This ageing of the global population has been described as the most important medical and social demographic issue worldwide [2]. Healthy ageing, defined as a process of maintaining functional ability to enable wellbeing in older age, sustaining mental and physical capacities and optimal psychosocial functioning, has been identified by WHO as a pressing strategic priority [2].

Robust evidence from an umbrella review of systematic reviews and meta-analyses demonstrates that physical activity improves physical function, reducing the risk of its age-related decline and preventing or delaying disease, disability, and mortality [3]. Physical activity is defined as any bodily movement that results in increased energy expenditure above resting metabolic rate [4] and is influenced by contextually situated interests, emotions, ideas, instructions, and relationships [5]. Physical inactivity, conversely, increases the risk of premature onset of ill health, disease, and frailty [6].

Levels of physical activity tend to decline throughout the life course, with older age groups having a higher proportion of individuals who are physically inactive or have low activity levels [7]. For example, it has been estimated that up to 50% of older adults are insufficiently active worldwide [8]. In Scotland, only 58% of adults aged 55–64 years, and 55% aged 65–74 years, currently meet the UK Chief Medical Officers' guidelines for moderate or vigorous physical activity (MVPA) [9].

Participation in physical activity is known to be influenced by social relationships. Indeed, studies have found that adults who have greater number of social contacts, more physically active social contacts, and close and reciprocal relationships, such as friendships, are more likely to be physically active [10]. This can, in-part, be explained by the Convoy Model of Social Relations, which theorises that individuals are surrounded by relationships that move with them throughout the life course, and that these relationships vary in their closeness, quality, structure (e.g., frequency of interaction, geographic proximity) and function (e.g., social support) [11]. Closer relationships are likely to remain stable throughout the life course, while more peripheral relationships are more dynamic and subject to change during life transitions such as retirement [12]. Furthermore, different types of networks can affect different health outcomes. For example, larger networks have been associated with better mobility outcomes, whereas smaller networks with high contact frequency have been associated with fewer disabilities [13]. Additionally, the theory of functional specificity asserts that different types of social contacts offer different types of resources, and people tend to selectively draw on specific ties depending on who is most likely to be

useful for a particular purpose [14]. For example, older adults may rely on spouses and close family for emotional needs, while friends provide opportunity for socialisation and physical activity [13].

"Co-engagement" in physical activity, people participating in physical activity together, may be a particularly viable avenue to increase physical activity as we age. Social network and social contagion theories [15] recognise that many individual health behaviours are influenced by the behaviours present within their social groups and relationships. That is, specific health behaviours become more likely when they are considered as 'normal' behaviours due to their prevalence in the wider network or population [16]. For example, research with other age groups has shown that adolescents will adopt levels of physical activity that are similar to their friends' physical activity levels over time [17]. To this end, family members' encouragement to engage in physical activity has been associated with increased co-engagement between parents and their adult children [18]. Within intervention settings, adults demonstrate larger increases in physical activity if their partner is also enrolled in the intervention [19] and, according to a recent systematic review [20], interventions centred on increasing physical activity may also concurrently improve social participation.

Interventions that leverage social contacts (i.e., 'social network interventions') to improve physical activity may do so by introducing new social contacts, or by aiming to enhance the social support and influence provided by existing social contacts. Social network interventions work by encouraging specific individuals to promote, model, and reward healthy behaviours to their social peers, which can influence social norms and raise self-efficacy [21]. Most such interventions attempt to use the existing social network to diffuse behaviour change, drawing upon existing social support, exchange, and influence processes, which may be less present in newer or more temporary social contacts [21]. Drawing on existing social contacts may therefore represent a viable leverage point for interventions to facilitate mutual co-engagement in activities, which have been demonstrated to be more effective than interventions based around one person providing social support to another, such as by providing information or encouragement about healthy behaviours [22]. The dual physical and social benefits of co-engagement are particularly important as people progress from mid-to-older age, as, in older adults, the effects of social activities may become more important and act as a buffer against the negative effects of ageing [23].

Accordingly, it is important to know which types of social contacts would be most beneficial to target in co-engagement interventions. This could depend on the characteristics of the person who is receiving the intervention, the characteristics of the social contact, or some characteristics of the relationship, such as emotional closeness, physical proximity, or frequency of contact. Previous research has mainly investigated how social network factors are associated with overall physical activity levels [10]. However, for the purposes of developing co-engagement interventions, it is essential to understand the specific characteristics that make pairs of individuals most likely to co-engage in physical activity, so that these types of relationship can be enhanced or facilitated as part of planned interventions. In addition, identifying barriers to engaging in physical activity with certain social contacts will help highlight areas for additional support to be put in place.

### Study aims

This study aimed to provide new insight into the types of social contacts that are important for mid-to-older aged adults' engagement in physical activity. Specifically, we sought to identify characteristics of participants, their social contacts, and aspects of their relationships that were associated with co-engagement. We hypothesised that older adults' co-engagement in physical activity varies across the sociodemographic characteristics of these adults and their social contacts, as well as the types of relationships and overall network structure. More specifically, we addressed the following research questions:

1. What are the key sociodemographic characteristics associated with mid-to-older aged adults' likelihood of co-engagement in physical activity with their social contacts?

2. Are mid-to-older aged adults more likely to co-engage in physical activity with social contacts who have specific attributes?

3. Is the nature of the relationship between mid-to-older aged adults and their social contacts associated with the likelihood of engaging in physical activity together?

4. Does the likelihood of mid-to-older aged adults' co-engagement with social contacts depend on the overall composition or structure of their social network?

## Materials and methods

### Data and participants

Data came from the Physical Activity, social ConnectednESs, and healthy ageing (PACES) study, which included 140 mid-to-older aged adults, aged 55–75 years (mean age = 65.6), who resided in the Renfrewshire and South Lanarkshire local authority areas in Scotland, United Kingdom. Although our average participant had not generally reached older ages where the age-related health outcomes related to physical activity levels would have fully manifested, we were interested in the preventative effects of co-engagement at an earlier stage, to inform the development of timely interventions [2]. The two purposively sampled areas were selected due to having a high proportion of people aged over 50 years, a high proportion of deprived areas, and a high proportion of the population aged over 65 years claiming pension credit, a means tested income support for pensioners on a low income [24]. These selection parameters were chosen because physical activity declines in adults aged over 55 years and is lowest in the most deprived quintile of area deprivation in Scotland [9]. Recruitment methods included mailing study flyers to potential participants in each of the areas (oversampling from postcodes in the lowest income decile), community partners digitally forwarding study information to individuals, and placing advertisements within partner organisations, community centres, libraries, and bus stops. Participants were also recruited via advertisements on social media.

Data collection was conducted from 25th August 2022–15th September 2023, either in person or remotely via screen-sharing capabilities on Zoom videoconferencing software. We note that this fieldwork period may have meant that participants were exposed to differences in temporal context that affected their physical activity, such as different weather or policy developments. The study was approved by the University of Glasgow Medical, Veterinary, and Life Sciences (MVLS) College Ethics Committee (Reference: 200210084). Informed written or verbal consent was captured by fieldworkers and recorded in a REDCap database. An egocentric social network design [25] was used for the survey (S1 Appendix). We used multiple name generators to capture the multidimensional nature of participant social networks and combined nominations into a single overall social network. Participants were asked to name the people with whom they had interacted in the last month, people they had not interacted with, but still considered themselves connected to, and which of these social contacts they were physically active with (see measures below). For contacts they were physically active with, there was also an option to select from a roster of names generated by the first two prompts. We opted not to limit the number of social contacts participants could name, in order to capture an appropriate amount of variation in who they interact with, allowing the inclusion of weaker social ties that may be overlooked if prompting for a fixed number of contacts [26]. They were then asked to provide details regarding the demographics of each contact (e.g., age, gender), as well as how they felt about the relationship (e.g., did they feel at ease with the person), and whether social contacts knew each other (specifically, whether they could have a conversation without the participant being present). The survey was conducted using Network Canvas, an open-source software for surveying networks [27].

### Sample

A total of 140 participants completed the survey, providing data on 3679 social contacts, an average of 26 social contacts each. As we used a multilevel modelling approach, our analysis is primarily conducted on the 3679 social contacts (level 1) nested within the 140 study participants (level 2). In multilevel analysis, statistical power is generally sufficient for

unbiased estimates of level 2 variation when the level 2 sample size is larger than 50 [28] and even lower samples can be sufficient for medium and large effect sizes [29].

## Materials and measures

This study investigated predictors of co-engagement in physical activity between mid-to-older aged adults and their social contacts. Predictors of co-engagement were measured across four domains, which are commonly used in multilevel social network analysis [30]: 1) individual factors, such as the participant's health or level of loneliness; 2) attributes of social contacts, such as age and gender; 3) attributes of the relationships, such as relationship type (e.g., family vs friend) and relationship quality; and 4) network measures, such as size. As such, conceptually distinct relationship attributes were used as both predictors and outcome, with the aim of assessing which types of relationship tend to make up the subset our participants were physically active with. Notably, in this style of social network survey, attributes of social contacts are based on the participant's perception of their social contacts rather than an objective measure (e.g., they might not know the exact age or health status of a social contact) [27]. However, the cognitive perception of the social network is of interest in itself and how people perceive their network can influence their own behaviour [31]. A full list of variables is included in Table 1. Missingness of variables ranged from 0 to 7.31% per variable.

## Outcome variable

**Co-engagement in physical activity.** Participants were asked: 'Are there people in your network whom you do physical activities with, like going for a walk, playing sports, going to an exercise class, or gardening?'. This 'name generator' prompt was designed to elicit social contacts with whom participants engaged in this specific type of interaction, similar to comparable studies of networks and physical activity [32] and tailored for our specific interest in co-engagement. Participants could assign previously nominated social contacts to this category or add additional social contacts that they recalled in response to this specific prompt, with all nominated social contacts combined as their social network and an additional binary variable indicating which of these they were active with. In this way, the outcome of the study is an attribute of the relationship between our participants and each social contact they nominated (in network terms, a 'tie-level' variable), with 3679 outcomes nested within 140 participants.

We selected co-engagement as our outcome variable as it captures a specific type of relationship which the literature suggests is important due to providing dual physical and social benefits [23]. Key outcomes of interest for the PACES study were physical activity and social connectedness. The outcome measure represents a point where these intersect and may represent useful levers for interventions [18]. This provides a new angle for thinking about physical activity, which is often measured as an individual attribute, rather than something embedded within relationships [10].

## Individual-level variables (participant characteristics)

Study participants were asked about a range of individual factors. *Age* was measured as a continuous variable, with values mean standardised to normalise the age range and distribution for inclusion in statistical models.

*Physical activity* was measured using the International Physical Activity Questionnaire (IPAQ). This measures three types of physical activity: low intensity activities, such as walking; moderate intensity activities, which require some effort and increase the rate of breathing; and vigorous intensity, which take hard physical effort and make people breath much harder than normal. The number of days per week and hours per day of each type of activity were recorded and used to calculate weekly energy expenditure, expressed as metabolic equivalent task (MET) minutes [33]. Participants with greater than three vigorous activity days and a MET of greater than 1500, or those with seven days of any type of activity and a MET greater than 3000, were classed as having high physical activity. Moderate physical activity was classified as

**Table 1. Sample description.**

| Domain | Variable | Mean | SD | Scale/ Proportion | Missingness |
|---|---|---|---|---|---|
| Outcome (n = 3679) | Physical activity co-engagement | | | 21.09% | 0 |
| Participant (n = 140) | Age | 65.58 | 5.57 | 55 years – 75 years | 0 |
| | Woman | | | 72.14% | 0 |
| | Retired | | | 73.57% | 0 |
| | Married | | | 60.0% | 0 |
| | Finances | 4.42 | 0.79 | 1 (difficult) – 5 (living comfortably) | 0 |
| | Dog owner | | | 22.14% | 0 |
| | Physical activity | | | High (n = 63, 45%); Moderate (n = 55, 39.3%); Low (n = 22, 15.7%) | 0 |
| | Health (eq5d utility) | 0.80 | 0.20 | 0 (worst health) – 1 (perfect health) | 1.43% |
| | Frailty | 1.44 | 0.71 | 0 (no frailty) – 4 (high frailty) | 0.71% |
| | Mental health (phq-4) | 3.61 | 0.74 | 1 (severe) – 4 (normal) | 0.71% |
| | Loneliness | 2.33 | 1.27 | 1 (never) – 5 (always or often) | 0 |
| Social contact (n = 3679) | Social contact age category | 5.05 | 1.78 | 1 (less than 20) – 8 (over 80) | 1.01% |
| | Same age category as participant | | | 33.0% | 1.01% |
| | Social contact gender - female | | | 62.5% | 0 |
| | Same gender as participant | | | 65.7% | 0 |
| | Social contact health | 4.06 | 0.98 | 1 (very poor) – 5 (very good) | 7.31% |
| | Social contact degree | 6.00 | 6.23 | Count | 0 |
| Relationship (n = 3679) | Frequency of contact | 3.11 | 1.60 | 1 (less than monthly) – 6 (daily) | 4.27% |
| | Lives close | 3.93 | 1.69 | 1 (not in the UK – 7 (live together) | 4.32% |
| | Length of time known | 4.24 | 1.21 | 1 (less than a year) – 5 (more than ten years) | 0.05% |
| | Relationship quality | 4.44 | 2.07 | 1 (low) – 7 (high) | 0% |
| | Relationship type | | | Friend (n = 1671, 45.4%); Family (n = 877, 23.8%); Other (n = 393, 10.7%); Child (n = 250, 6.8%); Neighbour (n = 211, 5.7%); Colleague (n = 154, 4.2%); Partner (n = 95, 2.6%); Parent (n = 26, 0.7%) | |
| Network (n = 140) | Size | 26.28 | 12.93 | Count | 0 |
| | Age External-Internal (EI) Index | -0.16 | 0.26 | -1 (all same age group) – 1 (all different age groups) | 0 |
| | Gender EI Index | -0.02 | 0.23 | -1 (all same gender) – 1 (all different genderl) | 0 |
| | Physical activity EI Index | -0.11 | 0.45 | -1 (all do PA with participant) – 1 (all don't do PA with participant) | 0 |

either three vigorous activity days amounting to over twenty minutes, at least thirty minutes of moderate activity and thirty minutes of walking, five or more moderate or walking activity days amounting to over thirty minutes, or at least five days of any activity amounting to at least 600 MET. Any participant not meeting these levels was classed as low physical activity. Including this as an independent variable allowed us to control for variation in co-engagement that may have been attributable to variation in individual physical activity levels.

*General health* was measured using the EQ-5D-5L [34], a standardised measure of health-related quality of life. All domains (self-care, mobility, usual activities, pain or discomfort, and anxiety or depression) were measured on a five-level scale, ranging from 1 (the most severe problems) to 5 (no problems) (Cronbach's alpha: 0.81). These were reverse coded so that higher scores reflected more severe problems, then an EQ-5D health utility score was calculated for each participant, using the EQ5DMAP command in Stata [35]. This produced scores with a potential range of 0 (death) to 1 (perfect health).

*Mental health* was measured using the Patient Health Questionnaire-4 (PHQ-4), a validated brief screening instrument for identifying anxiety and depression, which has established internal reliability, construct validity, and factorial validity [36]. The four PHQ-4 items were used: 'feeling nervous, anxious, or on edge', 'not being able to stop or control worrying', 'having little pleasure or interest in doing things', and 'feeling down, depressed, or hopeless', with responses ranging from 0 (not at all) to 3 (nearly every day). Scores were summed and represented as mental health categories: 'normal' (0–2), 'mild' (3–5), 'moderate' (6–8), or 'severe' (9–12). These four categories were converted to a four-point numeric scale with higher values represented better mental health (Cronbach's alpha: 0.85).

*Loneliness* was assessed with a single direct-item measure of loneliness. Participants were asked, 'how often do you feel lonely?', with responses ranging from 1 (never) to 5 (often or always). Measuring loneliness directly is an alternative to using an indirect measure, such as the UCLA Loneliness Scale [37]. Comparison of the two approaches has found a significant positive association between age and loneliness, and that older adults were more likely to report loneliness on the direct measure [37].

Due to evidence that *dog ownership* is associated with physical activity, health benefits, and increased social interaction [38], we included a binary (yes/no) measure of whether participants owned a dog as a control variable.

Participants also provided information on several demographic characteristics that were not retained in the statistical models due to sample homogeneity, high correlation with other variables, or were tested in initial modelling and found to not significantly contribute to the model. For example, we tested variables such as employment status, relationship status, and subjective financial situation.

## Attributes of social contacts

The *age of social contacts* was recorded as age categories (e.g., 'less than 20', '20-30', '40-50', with a maximum of 'older than 80' years). *Gender of social contacts* was recorded as man/woman/other and recoded as woman or other, due to the sample being majority women and only a single 'other' response. A binary *same age* variable and a binary *same gender* variable were created to identify when the social contact was in the same age or gender category as the participant. Participants were asked about the *physical health of social contacts*, with responses per contact ranging from 1 (very poor) – 5 (very good). The *degree* of each social contact (i.e., the number of connections they have to other social contacts, captured by asking if social contacts knew each other) was calculated as a measure of their centrality within the network. Highly connected social contacts may be able to diffuse health behaviours more effectively, as they tend to bridge groups and are interacted with in a greater variety of social contexts [39].

## Attributes of the relationship between participants and their social contacts

*Relationship type* described the connection between participants and social contacts, including friend, partner, child, colleague, family, neighbour, parent, and other. There was no free-text option to capture additional information on relationships described as other. Participants were also asked how *long they had known* each social contact, spanning from 1 (less than a year) - 5 (more than ten years). *Interaction frequency with social contact*, including contact in person, by phone, or via social media, ranged from 1 (less than once a month) to 6 (at least daily). Whether the social contact *lived close* ranged from 1 (they don't live in the UK) to 7 (we live together).

A measure of *emotional closeness* was included, ranging from 1 (not very close) to 5 (very close). Participants were also asked binary questions about whether they *felt at ease and could talk about personal and sensitive matters* with each person, and if they *felt close to the person and could call on them in times of need*. These three items were combined into a composite measure representing overall *relationship quality* with a range of 0 (lowest) to 7 (highest) (Cronbach's standardised alpha: 0.81).

## Measures of social network structure

*Network size* was calculated as the total number of social contacts that the participant mentioned, across all name generators. Outliers below or above a threshold of 1.5 times the interquartile range were removed, to manage any variation introduced by very large networks, and the measure was standardised by rescaling values within a range of 0–1. The structure of the network was captured by asking participants which of their social contacts knew each other. This allowed us to assess several measures of social network structure based on measuring these patterns of connections.

*The tendency of social contacts with the same attributes to be connected* was calculated using External-Internal (EI) index, which measures the relative homogeneity of connected groups of social ties based on a specific characteristic. We calculated this for *age category, gender*, and *whether they did physical activity with the participant*. This enabled us to capture whether there was clustering of social contacts with similar characteristics [40]. It results in a measure of -1 (complete homogeneity) to 1 (complete heterogeneity) on the selected attribute, with a negative EI indicating people with the attribute tend to all know each other, and a positive EI indicating that the attribute has no bearing on who knows each other in the network. Age and gender EI were included to control for general demographic-related homophily patterns that can occur in networks [41] and physical activity EI was included to detect whether participants had distinct groups of people to be active with. This assessment of whether co-engagement contacts tend to have ties to each other was sought to understand whether co-engagement tends to occur within specific pairs of individuals or broader groups, providing insight into the tendency for homophily and the types of intervention development that may be informed [21].

## Statistical analyses

We answered our research questions by testing a series of multilevel models [30]. The following modelling procedures were used. First, a variance components model, with co-engagement in physical activity as the outcome and no predictor variables, was tested to assess whether there were significant between-participant differences in co-engagement in physical activity. Then, we ran a series of multilevel models that tested groupings of predictors of co-engagement in physical activity. These were: the demographics and health of study participants (Model 1), attributes of social contacts (Model 2), attributes of the relationship between participants and social contacts (Model 3), and measures of social network structure (Model 4). All predictors were included in a final model (Model 5). These models generated Odds Ratios (OR) representing the association between specific characteristics and the probability of co-engagement [30].

We present bivariate correlations between key variables in supplementary materials (S2 Appendix). Due to a high level of correlation between several pairs of variables, the less theoretically relevant variables from correlated pairs were dropped from our models (e.g., a high correlation between measures of health and frailty led to the exclusion of frailty, since our health measure covered a broader range of health factors). Variance Inflation Factor (VIF) was calculated for all models to detect multicollinearity. All values were between 1–5, with the majority between 1–2, which suggests moderate correlation, but not severe enough to cause problems with model fit [42]. Bivariate analyses using Pearson's correlation coefficient confirmed only weak correlations between any combination of variables included in the models, including those measuring separate attributes of relationships, confirming that our outcome was a distinct construct from other relationship attributes used as predictors.

We also tested two random slope models to determine whether the association between relationship quality and co-engagement in physical activity, and the relationship between geographic proximity and co-engagement in physical activity, differed across participants. These parameters were chosen given that relationship-level measures are a key strength and novelty of our research design, and these two have the most potential for alteration and therefore relevance for social network interventions [39] in comparison to more static attributes such as the length of the relationship.

All analyses were conducted in RStudio Version 4.3.1. Multilevel analyses were conducted using the lme4 package [43]. Missing data were not imputed, as the amount of missing data was relatively low due to data collection being conducted by fieldworkers. Robustness checks were carried out using standardised measures of social contact degree and EI indices, with the direction and significance of all predictors found to remain constant in adjusted and unadjusted models.

## Results

### Sample description

The sample had a mean age of 65.6 years and was 72% female (Table 1). Furthermore, 60% percent were married and 73.6% were retired. The mean score for subjective financial status was towards the top end of the range (mean = 4.42, range = 1–5) representing a score that falls between 'doing alright' and 'living comfortably'. Across the entire sample, 21.1% of social contacts were identified as someone the participants co-engaged in physical activity with (min = 0%, 25th centile = 0.09%, median = 15.4%, 75th centile = 27.6%, max = 91.7%).

### Multilevel analysis

Our variance components model demonstrated that 19.7% of the variance in the likelihood of co-engagement in physical activity with social contacts was between participants. The results from the multilevel analyses are presented in Table 2.

Model 5 serves as the final model, in which all domains were tested together, while controlling for participant-level clustering. In this final model, none of the participant-level demographic and health attributes were associated with the probability of co-engagement in physical activity with social contacts. However, in the unadjusted Model 1, which tested this domain without controlling for information about social contacts, positive associations were found for participant age (OR = 1.16, 95% CI = 1.00, 1.35), high physical activity (OR = 1.41, 95% CI = 1.00, 1.98), better mental health (OR = 1.47, 95% CI = 1.08, 2.01) and dog ownership (OR = 1.49, 95% CI = 1.00, 2.22).

Several attributes of social contacts were significant predictors of co-engagement in the final model. Participants had 1.13 times higher odds of co-engaging in physical activity with social contacts for each one-unit increase in the age of social contacts (OR = 1.13, 95% CI = 1.04, 1.24), and 1.30 times higher odds with people in the same age category as themselves (OR = 1.30, 95% CI = 1.01, 1.67). The likelihood of co-engagement in physical activity was 1.69 times higher for each one-unit increase in the health of social contacts (OR = 1.69, 95% CI = 1.47, 1.94). Participants were more likely to engage in physical activity with social contacts who were more connected to their other social contacts, with each one-unit increase in degree associated with 1.03 higher odds of co-engagement (OR = 1.03, 95% CI = 1.01, 1.06). Gender homophily (participant and social contact being the same gender) was not significantly associated with co-engagement in the final model (OR = 1.25, 95% CI = 0.95, 1.66), but was significantly associated in the unadjusted Model 2, which only tested for social contact attributes (OR = 1.29, 95% CI = 1.03, 1.61).

Several relationship variables were significant in the final model. Compared to partners, participants were equally likely to co-engage in physical activity with friends (OR = 1.39, 95% CI = 0.71, 2.75) and significantly less likely to co-engage in physical activity with other family members (OR = 0.27, 95% CI = 0.14, 0.53), neighbours (OR = 0.11, 95% CI = 0.04, 0.28), or colleagues (OR = 0.06, 95% CI = 0.01, 0.22). They were more likely to co-engage in physical activity with people who lived closer (OR = 1.10, 95% CI = 1.01, 1.19), and those they interacted with more frequently (OR = 1.68, 95% CI = 1.53, 1.86), but less likely with people they had known for longer (OR = 0.66, 95% CI = 0.58, 0.75). Relationship quality was not

**Table 2. Results of multilevel models predicting co-engagement in physical activity.**

| Predictors | M1 Odds Ratios | CI | M2 Odds Ratios | CI | M3 Odds Ratios | CI | M4 Odds Ratios | CI | M5 Odds Ratios | CI |
|---|---|---|---|---|---|---|---|---|---|---|
| Intercept | 0.01 | 0.00–0.06 | 0.01 | 0.01–0.02 | 0.06 | 0.02–0.17 | 0.18 | 0.12–0.26 | 0.00 | 0.00–0.02 |
| Age | 1.16 | **1.00–1.35** | | | | | | | 1.04 | 0.87–1.25 |
| High physical activity | 1.41 | **1.00–1.98** | | | | | | | 1.27 | 0.84–1.90 |
| Physical Health | 2.27 | 0.79–6.50 | | | | | | | 1.90 | 0.54–6.67 |
| Mental Health | 1.47 | **1.08–2.01** | | | | | | | 1.16 | 0.81–1.67 |
| Loneliness | 1.12 | 0.96–1.30 | | | | | | | 1.14 | 0.96–1.36 |
| Dog Owner | 1.49 | **1.00–2.22** | | | | | | | 1.34 | 0.85–2.12 |
| Woman | 1.21 | 0.82–1.77 | | | | | | | 1.04 | 0.65–1.68 |
| Contact Age | | | 1.19 | **1.12–1.27** | | | | | 1.13 | **1.04–1.24** |
| Same Age | | | 1.32 | **1.08–1.62** | | | | | 1.30 | **1.01–1.67** |
| Contact Woman | | | 1.04 | 0.84–1.30 | | | | | 1.01 | 0.77–1.31 |
| Same Gender | | | 1.29 | **1.03–1.61** | | | | | 1.25 | 0.95–1.66 |
| Contact Health | | | 1.64 | **1.46–1.85** | | | | | 1.69 | **1.47–1.94** |
| Contact Degree | | | 1.06 | **1.05–1.08** | | | | | 1.03 | **1.01–1.06** |
| Child* | | | | | 0.57 | 0.31–1.02 | | | 0.61 | 0.30–1.23 |
| Colleague* | | | | | 0.06 | **0.02–0.21** | | | 0.06 | **0.01–0.22** |
| Family* | | | | | 0.25 | **0.14–0.44** | | | 0.27 | **0.14–0.53** |
| Friend* | | | | | 1.48 | 0.84–2.59 | | | 1.39 | 0.71–2.75 |
| Neighbour* | | | | | 0.09 | **0.04–0.21** | | | 0.11 | **0.04–0.28** |
| Other* | | | | | 0.69 | 0.35–1.38 | | | 0.83 | 0.37–1.88 |
| Parent* | | | | | 0.25 | 0.06–1.02 | | | 0.39 | 0.09–1.75 |
| Live Close | | | | | 1.18 | **1.10–1.27** | | | 1.10 | **1.01–1.19** |
| Relationship Length | | | | | 0.67 | **0.60–0.75** | | | 0.66 | **0.58–0.75** |
| Frequently Interact | | | | | 1.64 | **1.50–1.79** | | | 1.68 | **1.53–1.86** |
| Relationship Quality | | | | | 1.14 | **1.05–1.23** | | | 1.03 | 0.94–1.12 |
| Network Size | | | | | | | 1.84 | 0.81–4.19 | 1.05 | 0.34–3.30 |
| Age EI | | | | | | | 0.81 | 0.43–1.53 | 0.56 | 0.26–1.19 |
| Gender EI | | | | | | | 0.54 | 0.26–1.12 | 0.48 | 0.20–1.15 |
| Physical activity EI | | | | | | | 0.58 | **0.39–0.84** | 0.91 | 0.57–1.47 |
| **Random Effects** | | | | | | | | | | |
| σ² | 3.29 | | 3.29 | | 3.29 | | 3.29 | | 3.29 | |
| τ₀₀ | 0.62 .egoID | | 0.87 .egoID | | 1.14 .egoID | | 0.53 .egoID | | 0.70 .egoID | |
| ICC | 0.16 | | 0.21 | | 0.26 | | 0.14 | | 0.17 | |
| N | 137 .egoID | | 139 .egoID | | 139 .egoID | | 125 .egoID | | 122 .egoID | |
| Observations | 3574 | | 3377 | | 3516 | | 3338 | | 2861 | |
| Marginal R² / Conditional R² | 0.047/ 0.198 | | 0.094/ 0.284 | | 0.318/ 0.493 | | 0.023/ 0.159 | | 0.395/ 0.501 | |
| AIC | 3397.322 | | 3165.558 | | 2784.955 | | 3362.631 | | 2381.949 | |

*Reference category: Partner

Significant confidence intervals are indicated in bold.

significant in the final model (OR = 1.03, 95% CI = 0.94, 1.12). However, Model 3, unadjusted for other factors, demonstrated that each one-unit increase in relationship quality was associated with a 1.14 increase in the odds of co-engagement in physical activity (OR = 1.14, 95% CI = 1.05, 1.23).

There were no significant associations between measures of overall network structure and co-engagement in physical activity in the final model. Non-significant results were found for network size, age EI, gender EI, and physical activity EI (the tendency of social contacts with the same age/gender/co-engagement status to be segregated from those with different characteristics). However, in the unadjusted Model 4, which only tested these predictors, there was a significant negative effect for physical activity EI (OR = 0.58, 95% CI = 0.39, 0.84). The EI index for physical activity measured the extent to which social contacts who do physical activity with the participant tend to interact with each other. Each one-unit increase in the EI value (where higher values indicate greater integration) was associated with a 1.42 reduction in the probability of the participant doing physical activity with social contacts. This would suggest that subgroupings of social contacts who are physically active with the participant increases the probability of co-engagement in physical activity.

Results from random slope models indicated significant variation across participants for both effects (Likelihood ratio test (LRT) for relationship quality: 84.99; LRT for geographic proximity: 38.41). A negative intercept-slope covariance for relationship quality (-2.58) implied that participants with above average quantity of co-engagement social contacts also tended to have below average effects of relationship quality. This means that higher relationship quality has a stronger effect on co-engagement for participants with fewer social contacts to do physical activity with (Fig 1). A negative intercept-slope covariance for geographic proximity (-0.35) implied that participants with above average quantity of co-engagement social contacts also tended to have below average effects of geographic proximity. This means that participants with many co-engagement contacts tended to have greater variation in how close these contacts lived, while participants with fewer were more likely to co-engage in PA with people who lived close (Fig 2).

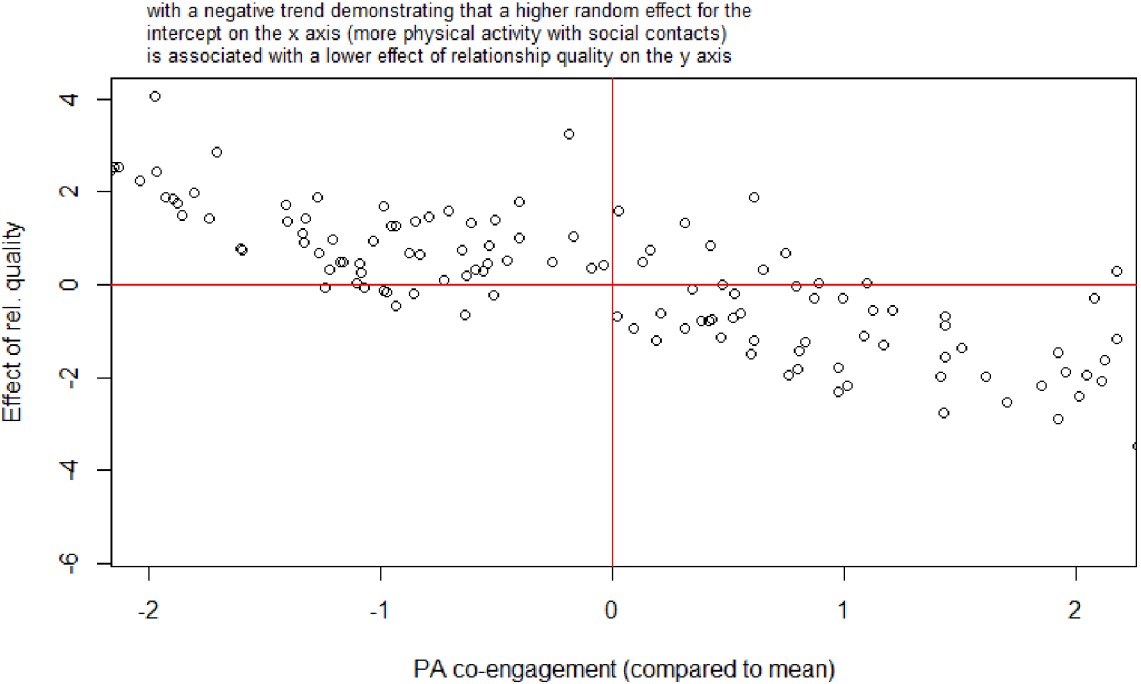

**Fig 1. Plot of slopes versus intercepts for relationship quality.**

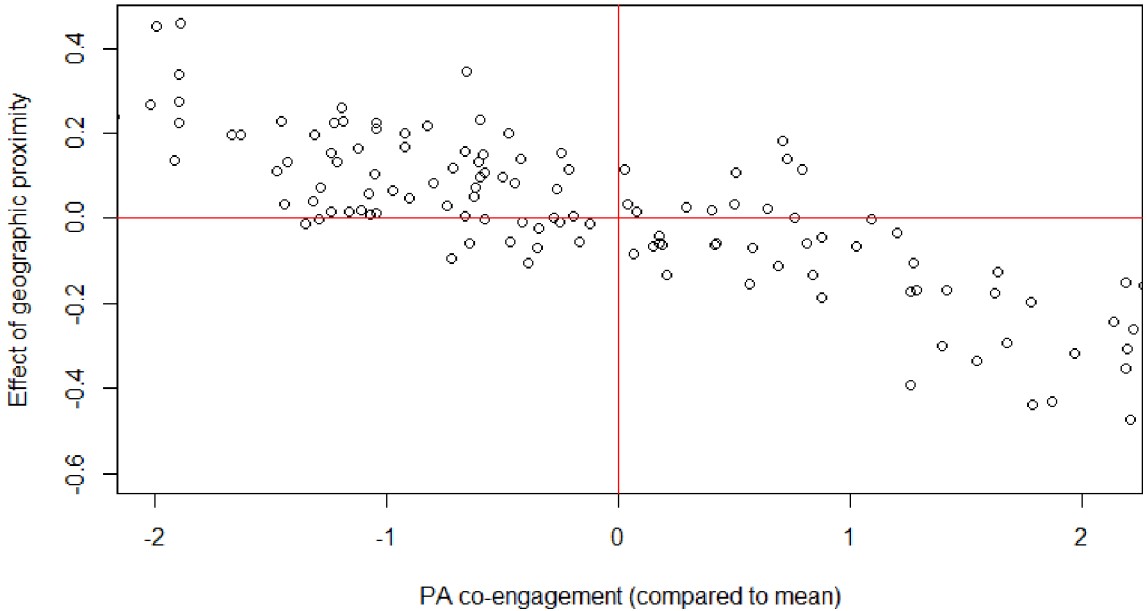

**Fig 2. Plot of slopes versus intercepts for geographic proximity.**

## Discussion

This study provides novel insights into the social dynamics around co-participation in physical activity in mid-to-late life. The findings have implications for health improvement, and the prevention of declining health in older age. In our sample, around 45% of participants aged 55–75 years had high physical activity levels, while around 16% reported low physical activity levels. Approximately 21% of all social contacts nominated by participants were people they co-engaged in physical activity with. We also found that there was significant variation between participants in the likelihood of doing physical activity with their social contacts. Co-engagement in physical activity was predicted by the attributes of specific social contacts, and characteristics of the relationship. They were more likely to co-engage with people the same age or closer to their own age, in good health, with more connections to their other social contacts, whom they considered as friends or partners, and whom they lived closer to and more frequently interacted with. They were less likely to co-engage with people they had known longer. There was no effect related to the individual attributes of participants, gender of social contacts, the quality of the relationship, or the number of social connections a person had. This suggests that, for co-engagement, the key factors were the accessibility and convenience of the appropriate types of social contact.

### Individual and network correlates of co-engagement in physical activity

No individual characteristics, or aggregated measures of their social networks, were significantly associated with co-engagement in the final model. Network size has been consistently associated with levels of individual physical activity in previous research [10]. However, in our study, network size was not significantly associated with the probability of co-engagement in physical activity. That is, having a greater number of people to potentially be active with did not lead to being active with a greater number of people. This suggests that co-engagement, as a phenomenon involving combined

physical activity and social interaction, may be associated with different factors than those associated with variation in overall physical activity levels of individuals.

In unadjusted models, the tendency of social contacts who do physical activity with the participant to be connected to each other [25] suggested that having interconnected physical activity partners increased the odds of co-engagement in physical activity. Network theory would suggest that this form of clustering may involve processes of social influence and social selection [39]. That is, people are more likely to be co-active due to the prevalence of social norms around co-engagement within their network [16], which can influence behaviours and raise self-efficacy that they can also engage in these behaviours [21]. Our findings show that these norms are not equally distributed across all social contacts, but clustered into groupings that have fewer connections to the rest of the social contacts. This is relevant, as group-related physical activity may have the additional benefits of improving mental health [44], reducing loneliness and depression [45], and improving social connectedness and social support [46]. Taken together, these factors may increase the likelihood that improvements in physical activity will be maintained long-term, although they were not significant when controlling for the specific attributes of social contacts and relationships. It is also notable that our unadjusted model found that high individual physical activity levels were associated with an increased likelihood of co-engagement, suggesting that co-engagement has not only social benefits [22] but benefits to overall activity levels similar to those found when people have support for physical activity [19].

## Social contact correlates of co-engagement in physical activity

The attributes of social contacts associated with co-engagement in physical activity, which held significance in the final model, were higher age, age homophily (being in the same age category as the participant), better physical health, and their centrality, as measured by the number of connections to other social contacts. Previous research has suggested that intergenerational physical activity interventions may be a promising solution to raise physical activity in different age groups, either by way of mixed-age interventions, such as dance classes [47] or the encouragement of paired physical activity between adults and children in the same family [18]. Our findings showed that adults aged 55–75 years were more likely to co-engage in physical activity with people in the same age range as themselves. Homophily, the tendency for people to connect to others with shared characteristics, is a recognised phenomenon in social network studies. For example, previous studies have identified the tendency for connections between people with similar levels of physical activity [10]. More generally, people tend to form and sustain connections with people based on a variety of shared attributes, due to possessing similar inclinations, beliefs, and perspectives [41]. Our study adds additional age-related context for older adults, showing a tendency for people of similar ages to engage in activities together. There may also be some tendency for homophily amongst people of the same gender, significant in our unadjusted model, which is relevant since gender inequalities have previously been associated with disparities in the use of physical activity spaces [48].

Furthermore, participants were more likely to co-engage with healthier social contacts. This suggests that people are more likely to engage in activities together when good health is a social norm within their network, or within a subgrouping of their network, which may in turn influence their own normative health behaviours [16]. Given that social contacts in more central network positions (more connections to the rest of the social contacts) were more likely to co-engage in physical activity, these individuals may be particularly well-placed to disseminate information and spread behaviour change to less physically active people in the network [39].

## Relationship correlates of co-engagement in physical activity

In terms of the relationship between participants and their social contacts, the type of relationship (friends and partners), the closer geographical proximity of the contact, a higher frequency of interaction, and a shorter length of time known were significant predictors of co-engagement. However, relationship quality (closeness, ability to count on, and feeling at ease with a person) was not significant when controlling for these factors. Previous research has shown that close and

reciprocal relationships are associated with higher levels of physical activity, because these types of relationship have the greatest influence on health behaviours [10]. Our findings suggest that the practical accessibility and convenience of social contacts are more important than the quality of the relationship for whether mid-to-older aged adults co-engage in physical activity. Moreover, our novel finding that they were less likely to do physical activity with people they had known for longer reinforces the finding that, for this age group, the convenience of relationships is more important than the duration (or quality) of the relationship. This provides a complementary insight to the current literature, which focuses more on the efficacy of different intervention designs (e.g., whether they focus on physical activity or social connectedness, whether they contain an educational component), rather than the specific attributes of relationships that may be important to guide intervention development [20].

Literature already exists on the physical convenience of places conducive to physical activity [49]. That is, people are more likely to walk for exercise or recreation when they have convenient places for exercise or recreation, such as neighbourhood streets, parks, and leisure facilities. Specifically, convenience has been conceptualised as proximity based on time and mode of transport (e.g., if the place can be accessed within 10 minutes walking). Walking is also influenced by factors such as health and having close friends to walk with [50]. Our findings suggest a complementary social phenomenon relating to the convenience of relationships. That is, older adults are more likely to co-engage in physical activity when they have accessible relationships who they live close to and frequently interact with. It is likely that having convenient social contacts to do activities with increases the salience and accessibility of co-engagement activities as a behavioural option, as well as influencing expectations of what constitutes normal or expected behaviour [51].

The relative importance of accessibility and convenience, as compared to individual factors, provides insight into the Convoy Model of Social Relations, which posits that social relationships are a key element of health and wellbeing, these relationships vary in quality, closeness, function, and structure, and can significantly exacerbate or buffer individual factors such as stress or socioeconomic status known to affect health [11]. Our findings exemplify this, specifically in the sense that social relationships influenced co-engagement in physical activity to a greater extent than individual factors (e.g., gender). Our study found that, for co-engagement in physical activity, the more objective characteristics of relationships, such as contact frequency or geographic proximity, were more relevant than their subjective closeness, quality, or function. Furthermore, the tendency to co-engage with newer relationships with no effect of relationship quality suggests that the co-engagement relationships are often those more dynamic, acquaintance-type outer levels of the social convoy that are lost and gained during age-related life transitions [12]. This can also be understood through the framework of functional specificity, which asserts that people engage with specific ties for specific purposes [52]. Our findings suggest that participants, particularly those with above average numbers of co-engagement ties, had subgroups of ties that they met for frequent interaction based around physical activity where the quality or strength of the tie was not as important as the ability to regularly interact.

## Implications

Our study identified the characteristics of social contacts and relationships associated with co-engagement in physical activity, identifying the types of people our participants were co-active with, without having received a physical activity intervention. We suggest that interventions may have an increased chance of success if they aim to facilitate and enhance activity between people with this naturally occurring tendency to co-engage in physical activity [21]. Findings suggest that interventions to promote co-engagement could utilise the homophily effect by connecting people who are in similar age groups. As such, it is important that interventions for this age group are age-appropriate, in terms of time, type, and intensity. Interventions aiming to encourage physical activity between existing social contacts could focus on encouraging friends and partners to be active together. There is less of a tendency for intergenerational co-activity or for mid-to-older aged adults to be co-active with neighbours or work colleagues, so this type of intervention may be an untapped resource. Mid-to-older aged adults who lack physically healthy social contacts may benefit from interventions to connect them with

healthier groups, or interventions that aim to influence social norms around healthy behaviour more broadly [21]. Alternatively, physical activity interventions may aim to provide supportive programmes for groups of individuals who are not physically healthy, where they can be provided with adequate support, adjustments, and understanding. Fundamentally, it is important that social contacts to do physical activity with are convenient and accessible.

## Strength and limitations

Our study successfully recruited 140 mid-to-older aged adults from two areas, who provided information on a total sample of over 3000 social contacts, to take part in a detailed social network survey that gathered substantial information on individual factors, such as their physical activity levels and various health measurements, as well as information on their social relationships. This approach generated a unique multilevel dataset, with information on social contacts nested within participant-level information. Previously, there have only been a small number of studies conducted exploring co-engagement in physical activity, and our use of a multilevel networks approach provided novel insight into characteristics that were associated with the likelihood of co-engagement in physical activity with different types of persons and relationships. We found strong statistical associations in sequential models and most of these were present in our final model. In the context of an ageing global population, findings such as the tendency to co-engage in physical activity within the same age group will provide valuable information for those involved in developing interventions and environments to promote physically active communities. Additionally, our multilevel modelling approach avoided the aggregation of network information into contextual-level proportions. Where an aggregated analysis may find an association between the number of friends in the network and higher physical activity levels, this does not necessarily indicate these friends are direct physical activity partners. To assume this can risk ecological fallacy, using observations about social context to draw conclusions about specific relationships [30]. We have modelled the odds of engaging in physical activity with each type of relationship in a more direct manner.

As a cross-sectional study, findings could not provide any insight into causal direction, meaning that any associations identified may flow in either direction or involve bidirectionality. Hence, we cannot perceive whether, for example, having healthier network members causes individuals to co-engage in physical activity with them, or whether people with a tendency to co-engage in physical activity will naturally have healthier people in their networks. In the context of intervention development, causal processes are important to demonstrate and difficult to untangle [39]. However, previous research has demonstrated that these causal processes do tend to explain much of the homogeneity found in social health behaviours [15]. Therefore, knowledge of the types of relationships associated with co-engagement in physical activity provide a starting point for further exploration. Second, our sample size meant that observations at the individual participant level may have lacked statistical power. This was not a problem for observations at the social contact level, as our 140 participants provided information on over 3,000 social contacts. However, it could explain the lack of significance for individual participant attributes in the final model. Our sample also had a lack of heterogeneity in terms of key demographics, such as income, meaning these were not included in the analysis, and our sample predominantly reported high incomes despite our efforts to sample from areas with higher levels of deprivation. Physical inactivity levels in our sample were lower, at 16%, than those in the Scottish Health Survey 2022 [9], which showed 42% (55–64 years) and 45% (65–74 years) did not meet moderate or vigorous activity levels. However, this is likely due to the fact that higher physical activity is also associated with higher education and income [33], which was reflected in our participant demographics. Furthermore, participants were interviewed over a one-year period in which their physical activity levels may have been affected by variation in environmental factors, although we did control for physical activity levels and found no significant effect on co-engagement in our final model. There was also a gender bias, with women overrepresented in our sample. This reflects a broader trend observed in studies relating to physical activity [53]. The Scottish Health Survey records that a lower proportion of women than men meet recommended physical activity levels, in the 55–64, 65–74, and over 70 age groups. However, our analysis controlled for gender of participants and contacts, with no significant effect. Finally, our

analysis did not model whether participants or their social contacts belonged to groups, clubs, or met at certain geographic settings. Future research may wish to further explore geographic factors that are associated with co-engagement in physical activity, social selection and influence processes, the role of community groups and clubs in shaping social network characteristics, the broader systemic influences of physical activity and social connectedness, the role of significant life-transitions that may occur in mid-to-older age, and to develop programme theories that support the development of interventions.

## Conclusions

Our results demonstrate that several relationship factors are associated with co-engagement in physical activity between mid-to-older aged adults and members of their social networks. Previous studies have also captured some of these factors, such as the importance of having friends to do physical activity with. However, some of our findings have diverged from previous studies, such as the finding that the size of the social network does not affect the likelihood of co-engagement in physical activity, but the frequency of interaction does. Our focus on co-engagement as the outcome differentiates this study from those that focus on predictors of individual physical activity levels, and our multilevel approach differentiates it from studies that measure social context at the aggregate level. This should be of particular relevance in the context of developing interventions that use co-engagement as a strategy to improve physical activity levels, as well as overall health and wellbeing. Key findings suggest that mid-to-older aged adults are more likely to co-engage with people of similar ages, and that the convenience of relationships matters more than the quality or quantity of relationships. We suggest that interventions aim to utilise these findings to promote co-engagement between people with the amenable attributes we have identified.

## Supporting information

**S1 Appendix. Protocol summary document.**
(PDF)

**S2 Appendix. Bivariate correlations of key variables.**
(DOCX)

## Acknowledgments

This research was conducted as part of the Physical Activity, Social Connectedness, and Healthy Ageing (PACES) study: www.gla.ac.uk/paces.

We would like to acknowledge the vital role of study fieldworkers, Jennifer Wheeler and Emma Smith, in study recruitment and data collection. We also thank Dr Elise Whitley for helpful comments on our first draft.

## Author contributions

**Conceptualization:** Mark McCann, Jelena Milicev, Kirstin Mitchell, Benjamin P Rigby, Sharon A Simpson, Emily Long.

**Data curation:** Martin Anderson, Emily Long.

**Formal analysis:** Martin Anderson, Emily Long.

**Funding acquisition:** Kirstin Mitchell, Sharon A Simpson, Emily Long.

**Methodology:** Martin Anderson, Srebrenka Letina, Mark McCann, Jelena Milicev, Grace Dibben, Abi MacDonald, Kirstin Mitchell, Laurence Moore, Jonathan R Olsen, Victoria J Palmer, Benjamin P Rigby, Sharon A Simpson, Meigan Thomson, Emily Long.

**Project administration:** Martin Anderson, Victoria J Palmer, Meigan Thomson, Emily Long.

**Supervision:** Kirstin Mitchell, Sharon A Simpson, Emily Long.

**Validation:** Srebrenka Letina, Mark McCann, Emily Long.

**Visualization:** Martin Anderson.

**Writing – original draft:** Martin Anderson.

**Writing – review & editing:** Martin Anderson, Srebrenka Letina, Mark McCann, Jelena Milicev, Grace Dibben, Abi MacDonald, Kirstin Mitchell, Laurence Moore, Jonathan R Olsen, Victoria J Palmer, Benjamin P Rigby, Sharon A Simpson, Meigan Thomson, Emily Long.

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
