## [Decision Letter · Decision Letter 0]

24 Sep 2024

PONE-D-24-30751Social network characteristics associated with mid-to older aged adults’ co-engagement in physical activityPLOS ONE

Dear Dr. Anderson,

Thank you for submitting your manuscript to PLOS ONE.  I have read and appreciated the article. After careful consideration, I feel that it has merit but does not fully meet PLOS ONE’s publication criteria as it currently stands. I ask you to address carefully the issues raised by the two reviewers and I invite you to submit a revised version of the manuscript. 

We look forward to receiving your revised manuscript.

Kind regards,

Mattia Vacchiano, Ph.D.

Academic Editor

PLOS ONE

Journal Requirements:

2.  Please include captions for your Supporting Information files at the end of your manuscript, and update any in-text citations to match accordingly. Please see our Supporting Information guidelines for more information: http://journals.plos.org/plosone/s/supporting-information .

Reviewers' comments:

Reviewer's Responses to Questions

**Comments to the Author**

1. Is the manuscript technically sound, and do the data support the conclusions?

Reviewer #1: Partly

Reviewer #2: Partly

2. Has the statistical analysis been performed appropriately and rigorously? 

Reviewer #1: Yes

Reviewer #2: Yes

3. Have the authors made all data underlying the findings in their manuscript fully available?

Reviewer #1: Yes

Reviewer #2: Yes

4. Is the manuscript presented in an intelligible fashion and written in standard English?

Reviewer #1: Yes

Reviewer #2: Yes

5. Review Comments to the Author

**Reviewer #1** : The paper adequately justifies the importance of studying older adults. It also recalls the importance of physical activity as a protective factor against morbidity and mortality. On the contrary, sedentary lifestyle becomes an important risk factor, especially for older people. In this context, it aims to explore what type of social contacts facilitate co-engagement in physical activities.

Although it refers to older people, the average age seems relatively low (66 years, which in some countries is close to the average retirement age). Although a decrease in physical activity is observed from the age of 55 onwards, the older population may be interesting to analyse precisely because of the impact it may have on their health.

The fieldwork lasted more than a year. This is a time in which significant changes in the environment can occur that affect mobility. It is worth remembering the ups and downs in social and physical activity linked to the COVID-19 pandemic, for example.

The average of 26 contacts is valuable, considering that a large part of the social support literature has focused on fewer than 10 contacts. This affects the structure of the personal network evaluated, as it is a dense core, which usually varies less between respondents. See for example: McCarty, C. (2002). Structure in personal networks. Journal of social structure, 3(1), 20. / Maya Jariego, I. (2018). Why name generators with a fixed number of alters may be a pragmatic option for personal network analysis. American journal of community psychology, 62(1-2), 233-238. / McCarty, C., Lubbers, M. J., Vacca, R., & Molina, J. L. (2019). Conducting personal network research: A practical guide. Guilford Publications. The application of multilevel analysis is also appropriate given the nature of the data.

There appears to be some overlap between the independent variables referring to relationships and the dependent variable (defined as "co-engagement"). The dependent variable refers not only to the performance of physical activities, but to the performance of physical activities jointly with some members of the network. Relationships are present in the predictor and the outcome. How is participation in physical activities (with other people) separated from the quality, quantity and structure of relationships with them?

On the other hand, selection and influence processes can be considered. Personal relationships facilitate physical activity, just as physical activity can facilitate the development of relationships. Why are the direction of relationships chosen over physical activity in relational contexts? Is the alternative explanation corresponding to the other direction of the relationship controlled in any way?

In that regard, the following conclusion seems confusing: "The insignificance of network size suggests that co-engagement may be driven by a different set of factors than individual physical activity, since network size has been consistently associated with individual activity levels in previous research".

The Methodology is explained in detail and clearly. However, there are some inconsistencies: In the Methodology section, it is stated that the mean age was 66 and, in the Results section, it is stated that it was 65.6. What was the real mean? On the other hand, the sample is clearly biased towards women (72 percent). Does this indicate a problem of accessibility or a refusal to participate by the male population? How does this affect the validity of the results?

It is interesting to note that relationships were not particularly relevant predictors of co-engagement. Could it be that the two variables are so closely related that the first does not serve to predict variability in the second? Also interesting is the predictive power of walking the dog, which is mentioned as an added circumstance in the Methodology section.

**Reviewer #2** : Thank you for the opportunity to review this study. This research aims to understand how co-engagement in physical activity may be related to the social networks of participants. It uses a multilevel approach with ego-alter tie at level 1 and ego characteristics (including the egonet properties) at level 2. This allows to go further than merely considering physical activity as an individual process, by including its interdependence with the persons present in the social surroundings of the individuals. The paper is well structured and the statistical analyses are well handled and performed, with the appropriate design to respond to the research questions. The results are well discussed and implications are aslo striking. However, a few complementary explanations and issues need to be addressed.

First, there lacks a precise description of the sample, in particular on how the different social domains (friends, family, colleagues, etc.) are distributed. Without such information, it is hard to tell if the subgroups' sample size are large enough to ensure sufficient statistical power.

I see that Physical activity is differentially operationalized whether it is the outcome variable or it is the physical activity of the respondent. The outcome variable is adapted from Mötteli and Dohle (2020) and the activity of the respondent is measured with the international IPAQ scale. Although one refers to co-engagement and the other refers to individual physical activity, I am wondering why the authors did not use a consistent measure.

It misses a rationale for the inclusion of key variables in the analyses. It is the case for the E-I indices and the degree of the alters, for instance. Although it is well discussed in the discussion section, elaborating on the importance of such variables in the literature section and providing hypotheses related to the research question would help understand why they are considered in the analyses.

The ‘Quality of the relation’ variable is a composite variable, which reaches a good internal consistency of .81. However, due to the different scales and ranges of the three variables (one variable uses a 5-point Likert scale and the two others are binary) some standardization seems to be needed before computing the composite score. Was this considered and can the authors provide a robustness check to see how this would impact this variable?

Density is included in the 'measure of social network' section but does not seem to be included in the models. Is there a reason for this? Is density only used to compute the E-I indices? Please explicit it as it is unclear.

From how the name generator procedure is described, it seems that participants are free to chose the number of alters they elicit. If that is the case, it results in unequal ego network sizes. This can have a potential impact on different variables included in the models as they are dependent on the size of the network. That is the case for degree centrality, as alters have a higher probability of having a higher degree in larger networks. The E-I index is also more likely to be smaller in larger networks. Although size can be considered as a control, are there any other ways the authors took it into account, and were there robustness checks with standardized measures for instance, or by hierarchically including the different variables?

6. PLOS authors have the option to publish the peer review history of their article (what does this mean? ). If published, this will include your full peer review and any attached files.

**Do you want your identity to be public for this peer review?** For information about this choice, including consent withdrawal, please see our Privacy Policy .

Reviewer #1: No

Reviewer #2: No

---

## [Author Response · Author response to Decision Letter 1]

1 Nov 2024

The authors thank you for the opportunity to submit a revised version of our manuscript, “Social network factors associated with mid-to-older aged adults’ co-engagement in physical activity” for publication in PLOS One. We appreciate your careful consideration and insightful feedback, which we have incorporated to make valuable improvements to the manuscript. We have attached a document titled "Response to Reviewers" which reponds to specific comments in detail.

---

## [Decision Letter · Decision Letter 1]

29 Jan 2025

PONE-D-24-30751R1Social network characteristics associated with mid-to-older aged adults’ co-engagement in physical activityPLOS ONE

Dear Dr. Anderson,

Thank you for submitting your manuscript to PLOS ONE. Following the review process of your manuscript, I am now requesting for minor revisions. This decision is the result of a second round of review and my own evaluation. One of the two reviewers provided positive feedback on your revisions, while the other declined to re-review the manuscript. Consequently, I sent the manuscript to a new anonymous reviewer, who has recently provided feedback. In light of this new feedback, I kindly ask you to address the points below of the new reviewer (hereafter **#reviewer 3** ). 

We look forward to receiving your revised manuscript.

Kind regards,

Mattia Vacchiano, Ph.D.

Academic Editor

PLOS ONE

Journal Requirements:

**Additional points to adress of the #reviewer 3:**

(1) Your results show that new relationships, particularly those that are not close, and the geographical proximity of alters with whom participants engage in physical activity are predictors of co-engagement. I was wondering if social participation, in terms of belonging to associations, clubs, etc., might be a hidden variable explaining co-engagement in physical activity. Clubs or associations tend to be local, bringing together people who live nearby but are not closely related. Additionally, they structure participation based on activity schedules and group people with similar characteristics (e.g., retired older adults, same age). Have you considered including membership in associations (e.g., walking clubs, sports clubs, etc.) in your research and model?

(2) In your paper, you have identified the types of social contacts positively associated with co-engagement. Conversely, are there any that are negatively associated with co-engagement? Are there certain types of social contacts that tend to hinder co-engagement in physical activity?

(3) Regarding types of relationships in networks, 11% are classified as "others". Do you have any additional information about this category? Who are they?

(4) In the "co-engagement in physical activity" section, you wrote: "Participants could assign previously nominated social contacts to this category or add new ones." Does this mean that with this second name generator based on physical activities, participants could extend their "interaction" network after using the first generator? Could you clarify this point?

(5) The degree of each social contact was calculated based on the connection "know each other" or "do physical activity with." This is not entirely clear to me.

(6) In Table 2, significant results are not marked with stars as usual but with confidence intervals in bold. It would be helpful to explain this format in advance for readers who may not be familiar with it. Additionally, it would be beneficial to recall your dependent variable in the first row of your table.

Reviewers' comments:

Reviewer's Responses to Questions

**Comments to the Author**

1. If the authors have adequately addressed your comments raised in a previous round of review and you feel that this manuscript is now acceptable for publication, you may indicate that here to bypass the “Comments to the Author” section, enter your conflict of interest statement in the “Confidential to Editor” section, and submit your "Accept" recommendation.

Reviewer #2: All comments have been addressed

2. Is the manuscript technically sound, and do the data support the conclusions?

Reviewer #2: Yes

3. Has the statistical analysis been performed appropriately and rigorously? 

Reviewer #2: Yes

4. Have the authors made all data underlying the findings in their manuscript fully available?

Reviewer #2: Yes

5. Is the manuscript presented in an intelligible fashion and written in standard English?

Reviewer #2: Yes

6. Review Comments to the Author

Reviewer #2: (No Response)

7. PLOS authors have the option to publish the peer review history of their article (what does this mean? ). If published, this will include your full peer review and any attached files.

**Do you want your identity to be public for this peer review?** For information about this choice, including consent withdrawal, please see our Privacy Policy .

Reviewer #2: No

---

## [Author Response · Author response to Decision Letter 2]

6 Feb 2025

Thank you to reviewer #3 for agreeing to reviewe our revised manuscript. We found your comments insightful and helpful for making some additional clarifications to the manuscript. We have attached a full response to each point in the uploaded 'Response to Reviewers' document.

---

## [Editor Report · Decision Letter 2]

12 Feb 2025

Social network characteristics associated with mid-to-older aged adults’ co-engagement in physical activity

PONE-D-24-30751R2

Dear Dr. Anderson, 

We’re pleased to inform you that your manuscript has been judged scientifically suitable for publication and will be formally accepted for publication once it meets all outstanding technical requirements.

Kind regards,

Mattia Vacchiano, Ph.D.

Senior Lecturer, University of Geneva

Academic Editor

PLOS ONE

---

## [Editor Report · Acceptance letter]

PONE-D-24-30751R2

PLOS ONE

Dear Dr. Anderson,

I'm pleased to inform you that your manuscript has been deemed suitable for publication in PLOS ONE. Congratulations! Your manuscript is now being handed over to our production team.

Kind regards,

on behalf of

Dr. Mattia Vacchiano

Academic Editor

PLOS ONE